# Transcriptome Analysis of Rice Reveals the lncRNA–mRNA Regulatory Network in Response to *Rice Black-Streaked Dwarf Virus* Infection

**DOI:** 10.3390/v12090951

**Published:** 2020-08-27

**Authors:** Tianze Zhang, Qian Liang, Chenyang Li, Shuai Fu, Jiban Kumar Kundu, Xueping Zhou, Jianxiang Wu

**Affiliations:** 1State Key Laboratory of Rice Biology, Institute of Biotechnology, Zhejiang University, Hangzhou 310058, China; 11416070@zju.edu.cn (T.Z.); chenyangli@zju.edu.cn (C.L.); fushuai@zju.edu.cn (S.F.); 2Key Laboratory of microorganism technology and bioinformatics research of Zhejiang Province, NMPA Key Laboratory for Testing and Risk Warning of Pharmaceutical Microbiology, Zhejiang TianKe High-Technology Development Co. Ltd., Hangzhou 310023, China; lq4977@tkgeneclub.com; 3Crop Research Institute, 16100 Prague 6, Czech Republic; jiban@vurv.cz; 4State Key Laboratory for Biology of Plant Diseases and Insect Pests, Institute of Plant Protection, Chinese Academy of Agricultural Sciences, Beijing 100193, China

**Keywords:** rice, rice black-streaked dwarf virus, transcriptome analysis, DElncRNA, DEmRNA, gene expression regulatory network, rice calli

## Abstract

The plant genome can produce long non-coding RNAs (lncRNAs), some of which have been identified as important regulators of gene expression. To better understand the response mechanism of rice plants to Rice black-streaked dwarf virus (RBSDV) infection, we performed a comparative transcriptome analysis between the RBSDV-infected and non-infected rice plants. A total of 1342 mRNAs and 22 lncRNAs were identified to be differentially expressed after RBSDV infection. Most differentially expressed transcripts involved in the plant–pathogen interaction pathway were upregulated after RBSDV infection, indicating the activation of rice defense response by RBSDV. A network of differentially expressed lncRNAs (DElncRNAs) and mRNAs (DEmRNAs) was then constructed. In this network, there are 56 plant–pathogen interaction-related DEmRNAs co-expressing with 20 DElncRNAs, suggesting these DElncRNAs and DEmRNAs may play essential roles in rice innate immunity against RBSDV. Moreover, some of the lncRNA–mRNA regulatory relationships were experimentally verified in rice calli by a quick and effective method established in this study. Three DElncRNAs were selected to be tested, and the results indicated that five mRNAs were found to be regulated by them. Together, we give a whole landscape of rice mRNAs and lncRNAs in response to RBSDV infection, and a feasible method to rapidly verify the lncRNA–mRNA regulatory relationship in rice.

## 1. Introduction

Rice black-streaked dwarf disease is an important rice disease in China, Japan, and many other Asian countries and is caused by Rice black-streaked dwarf virus (RBSDV) [1]. RBSDV is a member of the genus *Fijivirus*, family *Reoviridae* [2], and is transmitted by the small brown planthopper (SBPH, *Laodelphax striatellus* Fallén) in a persistent and circulative-propagative manner [2,3]. In addition to rice, RBSDV can also infect maize, wheat, and several species of weeds [2,3,4,5] to cause Maize rough dwarf disease and wheat dark-green dwarf disease [6,7]. RBSDV virions are non-enveloped, icosahedral, double-layered, 75–80 nm in diameter, and contain 10 double-stranded (ds) RNAs (designated as S1 to S10) with approximately 1.8 to 4.5 kilo-bases [1,8]. Microscopic analysis of ultrathin sections shows that RBSDV virions are restricted to the phloem tissues in infected plants, and the viroplasm, virus crystals, and tubular-like structures can be found in both infected plant and insect cells [9,10]. Several studies have shown that the expression of mRNAs and microRNAs in rice and maize plants can be affected by RBSDV infection. For example, the expression of rice signal transduction-associated genes can be altered after RBSDV infection, resulting in inhibition of the abscisic acid (ABA) pathway and induction of the jasmonic acid (JA) pathway [11,12]. In another study, many cell wall biosynthesis-related, chloroplast function-related, and disease resistance or stress-related genes were found to be significantly upregulated after RBSDV infection in maize [13].

Non-coding RNAs (ncRNAs) have been shown as important regulators of gene expression and can be divided into two groups: small ncRNAs and long ncRNAs (lncRNAs) [14,15]. Although the function of small ncRNAs has been relatively well-studied, there is still much to learn about lncRNAs, especially in plants. Recent transcriptome-wide studies using high-throughput RNA-seq have proved that the plant genome transcribes many lncRNAs. Additionally, many studies have indicated that lncRNAs are important in almost all plant biological processes [16,17]. However, only a few lncRNAs have been identified to regulate rice growth and development. For example, a long-day-specific male-fertility-associated RNA (*LDMAR*) was reported to influence rice anther development by regulating photoperiod-sensitive male sterility (PSMS) [18], and an *LRK* Antisense Intergenic RNA (*LAIR*) was reported to increase rice grain yield [19]. In addition, increasing evidence has shown that multiple lncRNAs have essential roles in plant resistance to abiotic or biotic stresses [20]. In *Arabidopsis thaliana*, a total of 1832 lncRNAs were found to be differentially altered under drought, cold, high-salt, and abscisic acid (ABA) treatments [21]. *Fusarium oxysporum* infection in *A. thaliana* also caused differential expression of many lncRNAs, and some lncRNAs were identified to play important roles in *A. thaliana* anti-fungal immunity [22]. Besides, ELF18-induced long-noncoding RNA1 (*ELENA1*) in *A. thaliana* has been found to interact with Mediator subunit 19a to regulate the expression of defense-related genes including *pathogenesis-related gene 1* (*PR1*) for enhancing the host resistance to *Pseudomonas syringe* infection [23]. However, only a few studies have been made to elucidate the roles of lncRNAs in plant-virus interaction. Gao and colleagues have discovered that *LINC-AP2* may be responsible for the formation of shorter stamen in the flowers of the Turnip crinkle virus (TCV)-infected *A. thaliana* plants, through anti-*cis* downregulation of *AP2* gene expression [24]. Moreover, a tomato lncRNA, *SlLNR1*, has been shown to directly interact with IR-derived vsRNAs from tomato yellow leaf curl virus (TYLCV) to inhibit TYLCV disease development [25]. To date, the expression profile and function of lncRNAs during virus infection in rice are still largely unknown.

The high-throughput RNA-seq technique has been successfully used to investigate the transcriptional profiles of different plants and to identify novel lncRNAs. To better understand the rice immune defense against RBSDV infection, and to identify novel lncRNAs involved in rice antiviral response, we analyzed the transcriptomic profile of mRNAs and lncRNAs in the RBSDV-infected and non-infected rice plants using an Illumina sequencing platform. Through this study, a total of 1342 differentially expressed mRNAs (DEmRNAs) and 22 differentially expressed lncRNAs (DElncRNA) in rice were identified after RBSDV infection. Analysis of the DElncRNA and DEmRNA co-expression network showed a complicated gene-regulating relationship, and then we focused on transcripts related to the plant–pathogen interaction pathway. To verify the regulatory relationship in this network, we silenced three DElncRNAs (TCONS_00065705, TCONS_00059822, and TCONS_00089479) in rice calli through *Agrobacterium*-mediated transformation of hairpin constructs and then detected the expression levels of predicted co-expressed mRNAs. The qRT-PCR result indicated that DEmRNAs (LOC_01g06730, LOC_06g44010, LOC_03g18030, LOC_03g10640, and LOC_04g51660) were, respectively, regulated by these three DElncRNAs. The data presented in this paper provide new clues to further investigate the response mechanism of rice to RBSDV infection and a feasible method to screen lncRNA-regulated mRNAs in rice.

## 2. Materials and Methods 

### 2.1. Plant Growth, Virus Inoculation, and RNA Isolation

Rice (*Oryza sativa*) cultivar Wuyujing No. 7 seedlings were grown inside an insect-free greenhouse at 28 °C and with a 16/8 h (light/dark) photoperiod. The rice seedlings were inoculated with RBSDV at 3-leaf-stage (3-L) using viruliferous small brown planthoppers (SBPH, *Laodelphax striatellus*) (five insects per seedling). The SBPHs were allowed to feed on the assayed seedlings for three days and were then removed. The inoculated seedlings were then grown inside individual containers in the greenhouse until the onset of disease phenotypes. Rice seedlings inoculated with non-viruliferous SBPHs were used as controls in this study. Leaf tissues were collected from individual assayed plants, frozen in liquid nitrogen, and stored separately at −80 °C until use. Leaf tissues harvested from a single plant were mixed and used as a biological replicate, and each treatment had three biological replicates. Total RNA was extracted from individual samples using TRIzol reagent (Invitrogen, Carlsbad, CA, USA), and virus infection in each RBSDV-inoculated sample was confirmed through RT-PCR using RBSDV-specific primers (F: 5′-GGTTGGAAAATGTTGAAAGCATTACA-3′ and R: 5′-AAACAAGGAGACACAAGATTGTAAAT-3′) targeting RBSDV segment 3.

### 2.2. Preparation of Transcriptome Libraries and Deep Sequencing

Total RNA (3 μg per sample) was used to construct each sequencing library. Briefly, ribosomal RNA was removed using the Epicentre Ribo-zero™ rRNA Removal Kit as instructed (Epicentre, Madison, WI, USA), followed by ethanol precipitation. The resulting mRNAs were fragmented using divalent cations under elevated temperature. The short mRNA fragments were then used to synthesize the first-strand cDNA using random hexamers primers. The second-strand cDNA synthesis was performed using DNA polymerase I and RNase H. The cDNA libraries were prepared by using the standard Illumina protocols and then sequenced on an Illumina Hiseq™ 4000 sequencing system (Illumina, San Diego, CA, USA) to generate reads with 150 bp paired-ends. Constructions of libraries and RNA sequencing were performed by the Zhejiang TianKe High-Technology Development Co., Ltd. (Hangzhou, Zhejiang, China).

### 2.3. Assembly and Annotation of Transcriptome Data 

First, quality-control filtering was performed by removing reads containing adapter or poly-N and the low-quality bases determined with the FastQC program, and high-quality clean reads were obtained. Then, reads derived from ribosomal RNAs were removed by mapping reads to the Ribosomal Database Project (http://rdp.cme.msu.edu/, accessed on 31 January 2018) with Bowtie2 v2.2.4 [26]. We obtained 41.637 Gb high-quality clean reads (6.939 Gb/per sample). All the subsequent analyses were conducted using high-quality clean reads. The resulting paired-end clean reads were aligned to the reference genome database using the TopHat v2.1.1 software (http://ccb.jhu.edu/software/tophat/index.shtml, accessed on 5 February 2018) as previously described [27]. In addition, Cufflinks v2.2.1 was used to assemble the mapped transcripts. The annotated transcripts were then used for further analyses. 

### 2.4. Identification of lncRNAs

The assembled transcriptome data were further screened to identify lncRNAs following four steps: (i) Cuffmerge software in the Cufflinks Support Package (v2.2.1) was used to combine all assembled transcripts, and those with less than three reads coverage were removed. (ii) Transcripts were removed with less than 200 bp in length. (iii) Cuffcompare software in the Cufflinks Support Package (v2.2.1) was used to compare the transcripts with annotated rice mRNAs (http://rice.plantbiology.msu.edu/, accessed on 20 February 2018), and the transcripts which were overlapped with the rice mRNAs were removed. (iv) Transcripts with potential coding capabilities were removed, which were assessed by three software, namely, CPC (Coding Potential Calculator, version 2), CNCI (Coding-Non-Coding Index, 0.9-r2), and Pfam 28.0 [28,29,30]. Finally, the remaining transcripts were identified as lncRNAs.

### 2.5. Differential Expression Analysis of mRNAs and lncRNAs 

The transcript abundances of mRNAs and lncRNAs were quantified using the normalized expression values as fragments per kilo-base per million reads (FPKM) by the Cuffdiff (v2.2.1) software [31]. Differentially expressed mRNAs (DEmRNAs) or lncRNAs (DElncRNAs) between RBSDV-infected and mock plants were selected using the threshold of |log2(fold change)| ≥ 1, and a statistical significance (false discovery rate (FDR) < 0.05). The expression-based sample clustering and the principal component analysis (PCA) were performed using the R package (v3.3.0) [32].

### 2.6. Analyses of GO and KEGG Enrichment

Gene ontology (GO) annotations were performed based on the Blast2GO database (http://www.blast2go.com, accessed on 1 March 2018), and the GO association was done using the BLASTX against the NCBI NR database. The GO enrichment analyses of DEmRNAs or the DElncRNA-targeted genes were performed using the topGO software at the Bioconductor (https://bioconductor.org/packages/release/bioc/html/topGO.html, accessed on 15 March 2018). The GO terms with corrected *p*-value < 0.05 were considered to be significantly enriched, and the three terms: biological process, molecular function, and cellular component, were defined with default parameters. Moreover, the metabolic pathway analysis of the DEmRNAs or the DElncRNA-targeted genes was performed using the KEGG database (http://www.genome.jp/kegg, accessed on 16 March 2018), and the statistical enrichments were determined using the phyper package in R package (v3.3.0). 

### 2.7. Prediction of DElncRNA-Targeted mRNAs and Construction of lncRNA–mRNA Regulatory Network

To analyze the DElncRNA *cis*-acting model, we searched genes in a region covering 100 kb up- to down-stream of individual DElncRNA. While the *trans*-acting model was analyzed using DElncRNAs and their co-expressed mRNAs. In this study, we calculated Pearson’s correlation coefficient (PCC) based on the expression level of each DElncRNA and mRNA pair with custom Perl scripts to search the common expression module. From individual DElncRNA and its *trans*-acting target mRNA, the pair with a PCC > 0.95 or <−0.95 was considered as significantly correlated expressed. The DElncRNA and DEmRNA co-expression networks were constructed according to the co-expression associations between DElncRNAs and DEmRNAs, and were visualized using the Cytoscape 3.2.7 software (http://cytoscape.org/, accessed on 17 September 2018).

### 2.8. Validation of DEmRNAs and DElncRNAs Using Quantitative Reverse Transcription-Polymerase Chain Reaction (qRT-PCR)

To validate the DEmRNAs and DElncRNAs revealed by the transcriptome data, leaves of RBSDV-infected and non-infected rice plants were collected. Total RNA (2 µg) isolated from individual leaf samples was used for cDNA synthesis using the ReverTra Ace qPCR RT Master Mix supplemented with a gRNA Remover as instructed (TOYOBO, Osaka, Japan). The resulting cDNA samples were individually diluted three times in RNase and DNase free water and then used (1.5 µL) in 10 µL qPCR reaction systems. Each qPCR reaction system consisted of 5 µL 2 × ChamQ Universal SYBR qPCR Master Mix (Vazyme Biotech, Nanjing, China) and 0.2 µL of each primer (10 mM), 1.5 µL diluted cDNA and 3.1 µL RNase and DNase-free water. The qPCR reaction was done on Light Cycler 480 II (Roche, Basel, Switzerland) set at 95 °C for 30 s, 40 cycles of 95 °C for 10 s, and 60 °C for 30 s. Melting curves were collected at 95 °C for 15 s, 60 °C for 1 min, 95 °C for 15 s. Each gene expression analysis was performed using three independent biological replicates. The expression of *OsUBC* (LOC_Os02g42314) was used as an internal control, and the relative expression of each gene was calculated using the 2^−∆∆CT^ method as described [33,34]. Differences were statistically evaluated with a *t*-test using GraphPad Prism 8. Primers used for DEmRNA and DElncRNA expression analyses are listed in Appendix A. 

### 2.9. Experimental Validation of the lncRNA–mRNA Regulatory Relationship

To construct the hairpin targeting DElncRNA, 300 to 600 bp forward and reverse sequences of the lncRNA, together with an intron from the *AtRTM1* gene of *Arabidopsis thaliana*, were fused into a pCAMBIA1300 vector containing the maize polyubiquitin gene promoter by the in-fusion cloning. The hairpin targeting the *gusA* gene of *Escherichia coli* was also constructed as a control. Then, the *Agrobacterium tumefaciens* strain EHA105 was transformed with recombinant plasmids containing hairpin constructs. The callus of rice cv. Wuyujing No. 7 was induced and then infected with the *Agrobacterium tumefaciens* harboring different hairpin constructs according to a previous protocol [35]. The 2N6 medium was used for callus induction; the 2N6-As medium with 200 mM acetosyringone was used for co-cultivation, and the AAM medium with 200 mM acetosyringone was used to suspend *Agrobacterium tumefaciens*. To increase the infection efficiency, we prolonged the soaking time of callus in *Agrobacterium tumefaciens* suspension to 30 min. After infection, the callus tissues were co-cultivated with *Agrobacterium tumefaciens* for three days. Then the *Agrobacterium tumefaciens* was killed by washing the callus tissues with 600 mg/L cefotaxime solution, and the callus tissues were continuously cultivated for four days before extracting total RNA for qRT-PCR analysis. The qRT-PCR method has been described above, and primers used for the detection of hairpin targeted lncRNAs and related mRNAs expression analyses are listed in Appendix A, primers used for vector construction are listed in Appendix A.

### 2.10. Data Availability

Raw reads of Illumina RNA-seq generated in this study are available from the Sequence Read Archive (SRA) at the National Center for Biotechnology Information (NCBI) under the project ID PRJNA657713 (https://www.ncbi.nlm.nih.gov/bioproject/PRJNA657713/, accessed on 19 August 2020). The transcriptome data generated in this study are available in Gene Expression Omnibus with series entry GSE156747 (https://www.ncbi.nlm.nih.gov/geo/query/acc.cgi?acc=GSE156747, accessed on 25 August 2020).

## 3. Results

### 3.1. RNA Sequencing Results

Three RBSDV-infected rice plants (e.g., RBSDV1, RBSDV2, and RBSDV3) and three non-infected rice plants (e.g., MOCK1, MOCK2, and MOCK3) were used in this study. A total of 299,865,814 raw reads were obtained through the Illumina sequencing. After quality-control filtering, 47,629,488, 46,285,766, 46,451,138, 45,328,332, 46,432,674 and 45,452,636 clean reads were obtained from the MOCK1, MOCK2, MOCK3, RBSDV1, RBSDV2 and RBSDV3, samples, respectively (Table 1). The Q20 and Q30 values of the clean reads were, respectively, over 96.15% and 89.85%, indicating that the obtained clean reads were of high quality (Table 1). Using the TopHat software v2.1.1, we mapped a total of 215,104,641 (71.73%) clean reads to the rice genome. Based on the characteristics of lncRNAs, a total of 1273 highly reliable lncRNAs were identified through four screening steps, and a total of 46,558 annotated mRNAs were also obtained and used in the subsequent analyses. 

### 3.2. RBSDV Infection-Induced Differential Expression Patterns of mRNAs and lncRNAs 

To compare the expression levels of mRNAs or lncRNAs between the RBSDV-infected and the mock plants, we normalized and analyzed the sequencing transcripts using the criteria of FDR < 0.05 and |log2(fold change)| ≥ 1 to identify DEmRNAs and DElncRNAs. The volcano plots showed the relationships between FDR and fold change of all the identified transcripts, representing the degree of difference and the statistical significance of gene expression between the RBSDV-infected and the mock plants (Figure 1a,b). We then generated heat maps using hierarchical clustering analyses of all the normalized DElncRNA values and the top 80 DEmRNAs determined by the R package software (Figure 1c,d). Hierarchical clustering analyses showed a systematic variation of mRNA or lncRNA expression between the RBSDV-infected and the mock plants. Furthermore, the principal component analysis (PCA) was performed to evaluate the sample distribution according to their expression profiles. The PCA charts showed that samples collected from the same treatment all gathered together (Appendix A). As a result, we identified a total of 1342 DEmRNAs, among which 1087 were upregulated, and 255 were downregulated after RBSDV infection (Figure 1e and Appendix A). Among these DEmRNAs, three genes (LOC_Os01g72370, LOC_Os01g03360, and LOC_Os06g36950) were the most affected DEmRNAs. LOC_Os01g72370, encoding a helix-loop-helix transcription factor related to the cellular response to iron ion starvation, and LOC_Os01g03360, encoding a Bowman-Birk type bran trypsin inhibitor (BBTI5) which participates in negative regulation of endopeptidase activity, were upregulated by 8.9-fold. LOC_Os06g36950, encoding a hobo/Ac/Tam (hAT) transposase, was downregulated by sixfold after RBSDV infection. At the same time, a total of 22 lncRNAs were identified to be differentially expressed. Among them, 17 lncRNAs were upregulated, and the other five lncRNAs were downregulated in the RBSDV-infected plants compared with those in the mock plants (Figure 1f and Table 2). A lncRNA (TCONS_00081336) showed the most significant upregulation (7.9-fold), while another lncRNA (TCONS_00074370) showed the most significant downregulation (9.2-fold). However, we could not predict their functions directly.

Although lncRNAs have been investigated for years, many of them have not been well annotated. To date, multiple lncRNAs have been considered to regulate gene expression and to shape 3D nuclear organization [36]. Using DElncRNAs identified in the RBSDV-infected rice plants, we investigated their general signatures, including their distributions on chromosomes and their length. The results showed that these DElncRNAs were distributed on almost all chromosomes of rice, except chromosomes 10 and 11. Among these DElncRNAs, the upregulated DElncRNAs were found on chromosomes 1, 4, 5, 7–9, while the downregulated DElncRNAs were found on chromosomes 2, 3, 6, and 12 (Figure 2a and Table 2). In contrast to DElncRNAs, the DEmRNAs were found widely distributed on all chromosomes, and more DEmRNAs were distributed on chromosome 1 (Figure 2b and Appendix A). The length of DElncRNAs ranged from 430 to 12,346 bp, with the majority between 3000 and 6000 bp (Table 2). 

### 3.3. Functional Characteristics of DEmRNAs

Based on the GO classification, the DEmRNAs can be separated into three categories: molecular function, cellular component, and biological process. In the molecular function category, DEmRNAs with oxidoreductase activity (GO: 0016491), iron ion binding (GO: 0005506), transferase activity and transferring hexosyl groups (GO: 0016758), and chitinase activity (GO: 0004568) were significantly enriched in the RBSDV-infected plants (Figure 3a,b). Because chitinase has been reported as one of the main plant disease-related proteins (i.e., PR proteins) [37], we considered that the upregulated expression (~4.5-fold) of genes with chitinase activities (LOC_Os11g47600 and LOC_Os04g41620) were related to rice resistance against RBSDV infection. Reactive oxygen species (ROS) plays a multitude of roles in rice defense response to RBSDV infection [12]. A lot of DEmRNAs were classified into the oxidoreductase activity (GO: 0016491), suggesting the regulation of cell redox status in response to RBSDV infection. In the cellular component category, DEmRNAs related to the integral component of membrane (GO: 0016021), cytoplasmic membrane-bounded vesicle (GO: 0016023), and plasma membrane (GO: 0005886) were significantly enriched. These genes may be important in vesicle formation during RBSDV infection in cells (Figure 3a). In the biological process category, DEmRNAs involved in the single-organism process (GO: 0044699) were significantly enriched, and DEmRNAs involved in the oxidation-reduction process (GO: 0055114), small molecule biosynthetic process (GO: 0044283), and organic acid metabolic process (GO: 0006082) were significantly upregulated upon RBSDV infection. Besides, DEmRNAs in the polysaccharide metabolic process (GO: 0005976), defense response (GO: 0006952), and transmembrane transport (GO: 0055085) were also affected (Figure 3a).

To obtain more information on the molecular and biological responses to RBSDV infection, we searched the KEGG database using the RBSDV-induced DEmRNAs, and found 871 DEmRNAs associated with KEGG pathways. The top 20 significantly enriched pathways are shown in Figure 3c. Among these 20 KEGG pathways, the metabolic pathway (ko01100) was the most enriched pathway with 274 DEmRNAs followed by the biosynthesis of secondary metabolites (ko01110) with 208 DEmRNAs, and then, the plant–pathogen interaction (ko004626) with 115 DEmRNAs (Figure 3c and Appendix A). In this study, the DEmRNAs associated with the plant–pathogen interaction pathway were significantly enriched with 86 upregulated DEmRNAs and 29 downregulated DEmRNAs, indicating the activation of this pathway by RBSDV infection (Appendix A). The gene encoding PR1 protein (LOC_Os01g28500), an important defense response protein, was upregulated by 6.6-fold, suggesting that the salicylic acid pathway might be activated after RBSDV infection. The gene encoding a respiratory burst oxidase (LOC_Os04g48930) was upregulated by fivefold, but the gene encoding an ABA-responsive element binding factor (LOC_Os02g03960) was downregulated by 2.1-fold, suggesting that the abscisic acid pathway might be suppressed during RBSDV infection. Previous studies demonstrated that plant hormones played important roles in plant defense against RBSDV infection [11,12,38]. Jasmonic acid and auxin signaling pathways enhance the resistance of rice to RBSDV, while abscisic acid and brassinosteroid pathways mediate the susceptibility to RBSDV infection [11,12,38]. In our results, we found that the DEmRNAs were also enriched in the plant hormone signal transduction pathway (ko04075), indicating broad responses of plant hormone signaling pathways to RBSDV infection. Other genes involved in the phenylalanine, tyrosine and tryptophan biosynthesis (ko00400), phenylpropanoid biosynthesis (ko00940), flavonoid biosynthesis (ko00941), and diterpenoid biosynthesis (ko00904) were also significantly enriched (Figure 3c and Appendix A). The gene encoding chalcone synthase (LOC_Os11g32650) in the flavonoid biosynthesis pathway, was upregulated by 5.6-fold. These results indicate that a complex gene expression network is affected by RBSDV infection, and defense response pathways are activated.

### 3.4. Functional Analyses of DElncRNA-Targeted Genes

*Cis*-acting lncRNAs are known to repress or activate the expression of genes that are adjacent to the transcription sites of lncRNAs on the same chromosome. It is also reported that *trans*-acting lncRNAs regulate gene expression through a variety of transcriptional or post-transcriptional biological processes, while *cis*-acting lncRNAs regulate the expression of nearby genes epigenetically at the transcriptional level [39]. To elucidate the function model of *cis*-acting lncRNAs, we searched mRNAs adjacent to the transcription sites of DElncRNAs. In this study, we searched 100 kb sequence up- to down-stream of the 22 identified DElncRNAs for their *cis*-regulatory effects and defined the relationship between the lncRNA and its adjacent mRNA as co-located. The result showed that a total of 304 mRNAs were co-located to the 22 DElncRNAs, suggesting that the expression of these 304 mRNAs might be affected by DElncRNAs through their *cis*-regulatory activities (Appendix A). GO categorization analyses of the predicted targets of these DElncRNAs showed that they were involved in a broad range of processes (Appendix A). In total, 109 targets were mapped into the KEGG pathway database and the top 20 enriched pathways are shown in Figure 4a and Appendix A. These results indicated that the co-located mRNAs involved in the plant–pathogen interaction and the ubiquitin–proteasome protein degradation pathway were more enriched compared with other pathways. Many studies have shown that the ubiquitin–proteasome protein degradation pathway plays important roles in the degradation of viral proteins. In contrast, viruses can also hijack plant ubiquitin–proteasome machinery to help their infections. Recently, He and colleagues reported that the P5-1 protein encoded by RBSDV could regulate the ubiquitination activity of SCF E3 ligase to inhibit the jasmonate signaling pathway to promote RBSDV infection [40]. In this study we also noticed that five mRNAs in the ubiquitin-mediated proteolysis (ko04120) were co-located with DElncRNAs (Figure 4a and Appendix A), suggesting that RBSDV infection in rice can regulate several genes related to the ubiquitin–proteasome pathway through lncRNAs. 

Among the mRNAs that co-located with DElncRNAs, 12 were differentially expressed, resulting in 12 co-located pairs of DElncRNA–DEmRNA in the RBSDV-infected plants (Table 3), and we speculate that these 12 DEmRNAs might be regulated by these 12 DElncRNAs, respectively. A gene that encodes a member in the transferase family (LOC_Os04g51660) was found to co-locate with TCONS_00059822 and involved in the flavonoid biosynthesis pathway, which was reported to play important roles in plant defense against pathogens and insects [41]. Thus, this finding suggested that TCONS_00059822 may regulate host flavonoid biosynthesis pathway through the *cis*-regulation of LOC_Os04g51660 by 1.79-fold. A previous report has indicated that a cytochrome *P450* gene is significantly upregulated in RBSDV-infected maize plants [13]. In this study, we have also observed a 3.9-fold upregulated rice cytochrome *P450* gene (LOC_Os01g72270) in the co-location data (Figure 4a and Table 3). We speculate that this cytochrome *P450* gene is likely regulated by TCONS_00011723 in the RBSDV-infected plants. In general, the target DEmRNAs were upregulated by 1.6- to 3.9-fold by the corresponding DElncRNAs (Table 3).

Next, we calculated the Pearson’s correlation coefficient (PCC) based on the expression level of each DElncRNA and mRNA pair, and the relationship of the DElncRNA–mRNA pair with a PCC > 0.95 or < −0.95 was defined as co-expressed. We analyzed the co-expressed relationships between DElncRNAs and mRNAs to predict the roles of DElncRNAs in the *trans*-acting model. The results showed a total of 17,335 matched co-expressed pairs (Appendix A). The GO categories of the matched mRNAs are shown in Appendix A. The KEGG analysis result demonstrated that the homologous recombination (ko03440) and the plant–pathogen interaction (ko04626) pathways were significantly enriched (Figure 4b and Appendix A). In these 17,335 co-expressed pairs, we filtered out 684 DEmRNAs as the potential targets of 22 DElncRNAs to investigate further (Appendix A). 

### 3.5. DElncRNA–DEmRNA Co-Expression Network

To predict the functions of DElncRNAs on DEmRNAs, we constructed the DElncRNA–DEmRNA co-expression network by using the 684 DElncRNA–DEmRNA co-expression pairs mentioned above (Figure 5 and Appendix A). This network contains 15 upregulated and 7 downregulated DElnRNAs, and 579 upregulated and 105 downregulated target DEmRNAs. It could be seen that the expression of the DElncRNA–DEmRNA pairs were positively correlated (Figure 5). In the network, three DElncRNAs (TCONS_00065705, TCONS_00089479, and TCONS_00056787) co-expressed with a large number of DEmRNAs, suggesting that they might play important roles during RBSDV infection (Figure 5 and Appendix A).

Previous studies have indicated that the plant–pathogen interaction pathway including the MAPK signaling pathway and the plant hormone signal transduction pathway participates in plant antiviral resistance. In this network, 20 DElncRNAs were found to be significantly and intricately interconnected with 56 DEmRNAs in the plant–pathogen interaction pathway, and the predicted co-expression network of these DElncRNA–DEmRNA pairs was also constructed (Figure 6 and Appendix A). In this network, expression levels of most DElncRNA–DEmRNA pairs were upregulated after RBSDV infection, suggesting the activation of the plant defense system. Additionally, we found that some genes could be simultaneously targeted by different DElncRNAs, implying the essential roles of these genes and lncRNAs in RBSDV infection. For example, three genes (LOC_Os11g10770, LOC_Os04g43440, and LOC_Os09g34160) were found to be co-expressed with seven, six, and five DElncRNAs, respectively (Figure 6 and Appendix A). Calcium (Ca^2+^) is an important messenger involved in plant responses, and Ca^2+^ concentration in the cytoplasm increases as one of the early signals of pathogen invasion [42]. In this co-expression network, two genes (LOC_Os08g27170 and LOC_Os01g04280), predicted to encode calmodulin-binding proteins, were co-expressed with five and four DElncRNAs, respectively (Figure 6 and Appendix A). In addition, *OsCML16* (LOC_Os01g04330) and *OsCML21* (LOC_Os05g24780) encoding the plant-specific calmodulin-like proteins, were found to be co-expressed with four DElncRNAs (TCONS_00010868, TCONS_00059316, TCONS_00056787 and TCONS_00059822) (Figure 6 and Appendix A). Jasmonic acid (JA) pathway is known to positively regulate plant defense against RBSDV infection [38]. In this network, *OsJAZ8* (LOC_Os09g26780) and *OsJAZ11* (LOC_Os03g08320), predicted to encode the key co-receptors of JA pathway, were both found to be co-expressed with two DElncRNAs (TCONS_00089479 and TCONS_00065705), indicating that the two lncRNAs might participate in the regulation of JA pathway. We also noticed that LOC_Os06g44010 which encodes *OsWRKY28* was also co-expressed with lncRNA TCONS_00065705 in this network. The WRKY gene family is a large transcription factor (TFs) family in higher plants, which is involved in various plant processes in coping with diverse biotic and abiotic stresses [43]. We found mRNAs related to the flavonoid biosynthesis pathway, such as LOC_Os03g18030 and LOC_Os10g16974. Taking the finding in the co-location analysis into account, we suspect flavonoids may be important compounds in the response of rice to RBSDV infection. These complex networks can provide effective evidence for estimating lncRNA function and clues for further study of the immune response induced by RBSDV invasion. 

### 3.6. Validation of DEmRNA and DElncRNA in Response to Viral Infection

To validate the Illumina sequencing data of the rice transcriptome and the expression patterns of the DEmRNAs and DElncRNAs, nine DEmRNAs and six DElncRNAs related to the plant–pathogen interaction pathway were selected for qRT-PCR analyses. The *OsUBC* gene was selected as an internal reference for the relative quantification as its expression level changes little upon RBSDV infection [33]. The selected nine DEmRNAs included upregulated LOC_Os01g06730, LOC_Os10g25090, LOC_Os01g28500, LOC_Os01g56330 and LOC_Os05g39720 as well as downregulated LOC_Os12g02450, LOC_Os01g23970, LOC_Os08g42700 and LOC_Os02g03960 in RBSDV-infected rice plants, and the qRT-PCR results were consistent with the Illumina sequencing data (Figure 7a). Besides, six randomly selected DElncRNAs including upregulated TCONS_00005781, TCONS_00056787, TCONS_00089479, TCONS_00008316, and downregulated TCONS_00040997, TCONS_00049628 were also validated by qRT-PCR (Figure 7b). Thus, these data together suggest that the transcriptome results are reliable and can represent the virtual expression pattern upon RBSDV infection. 

### 3.7. Experimental Validation of the lncRNA–mRNA Regulatory Relationship in the DElncRNA–DEmRNA Co-Expression Network

To experimentally confirm the regulatory relationship between lncRNA and mRNA in the DElncRNA–DEmRNA co-expression network, and to screen lncRNAs and mRNAs for future study, we attempted to detect the expression levels of DEmRNAs in their co-expressed DElncRNAs-silenced rice calli. Three DElncRNAs (TCONS_00065705, TCONS_00089479, and TCONS_00059822), and eight mRNAs were selected from the 20 DElncRNAs and 56 DEmRNAs mentioned above for further tests (Table 4). To silence these three lncRNAs, the hairpin construct targeting each lncRNA was expressed in rice calli by *Agrobacterium*-mediated transformation, while the gus-hairpin construct was expressed as a negative control. The qRT-PCR analyses showed that these three lncRNAs were effectively silenced at 7 days after *Agrobacterium* infection (Figure 8a–c). In five selected mRNAs (LOC_Os01g06730, LOC_Os06g44010, LOC_Os03g18030, LOC_Os01g28500, and LOC_Os03g03034) predicted to be co-expressed with TCONS_00065705, the relative expression levels of LOC_Os01g06730, LOC_Os06g44010, and LOC_Os03g18030, which, respectively, encode a receptor-like protein, a WRKY transcription factor family protein OsWRKY28, and a leucocyanidin oxygenase, were significantly reduced after silencing TCONS_00065705 (Figure 8d and Table 4). Both LOC_Os04g51660 encoding a malonyl transferase and LOC_Os03g10640 encoding a calcium-transporting ATPase were significantly downregulated after silencing TCONS_00059822 (Figure 8e and Table 4). Interestingly, we found that the expression of LOC_Os01g06730 was also inhibited in TCONS_00089479-silenced rice calli (Figure 8f and Table 4), which suggested that LOC_Os01g06730 was regulated by the two lncRNAs (TCONS_00065705 and TCONS_00089479) and might play an important role in rice-RBSDV interaction. Whereas, there were some mRNAs that their expression levels had no significant changes in co-expressed lncRNAs-silenced rice calli (Appendix A).

## 4. Discussion

Non-coding RNAs (ncRNAs) are RNAs that do not encode proteins and can be divided into two groups by their size: (*i*) small non-coding RNA, including microRNAs (miRNAs), small interfering RNAs (siRNAs), piwi-interacting RNAs (piRNAs), *trans*-acting siRNAs (ta-siRNAs) and natural antisense transcript siRNAs (NAT-siRNAs), and (*ii*) long non-coding RNAs (lncRNAs), which are longer than 200 nt [44]. Earlier studies have suggested that many of the lncRNAs and mRNAs may share the common mechanisms of transcription and processing [45]. It is noteworthy that the lncRNAs are more cell-type specific, less abundant, and conserved compared with mRNAs. Moreover, some lncRNAs have been identified diversified and essential functions in gene silencing and imprinting, transcription, mRNA splicing, translation, trafficking of nuclear factors, genome rearrangements, and regulation of chromatin modifications [46]. However, whether lncRNAs have specific biological functions or are merely transcriptional by-products remains controversial, because most of them do not have a recognizable function. Current studies on plant lncRNAs are less comprehensive than those for other eukaryotes [45]. Although multiple lncRNAs have now been identified in Arabidopsis [21], rice [47], maize [48], cotton [49] and Medicago [50], and have been shown to play roles in plant vegetative growth and reproductive development, very few studies focused on plant lncRNA functions in the gene regulatory networks associated with plant defense responses [18].

Several earlier studies have investigated the expression of mRNAs and microRNAs in the RBSDV-infected maize and rice plants and have found some molecular pathways involved in maize-RBSDV and rice-RBSDV interactions [12,13,38,51]. However, the roles of lncRNAs in RBSDV infection are still unclear. To investigate how RBSDV infection affects lncRNA expression in rice, we analyzed the expression of lncRNAs and mRNAs in RBSDV-infected and non-infected rice plants through RNA-seq. In this study, a total of 1342 DEmRNAs have been identified. Among these DEmRNAs, 1087 were upregulated and 255 were downregulated in RBSDV-infected plants (Figure 1e). Moreover, we have also identified 1273 lncRNAs. Among them, 17 were upregulated and five were downregulated in RBSDV-infected rice plants (Figure 1f). GO enrichment and KEGG enrichment analyses for DEmRNAs showed that, similar to previous findings, RBSDV infection in rice also altered the expression of many genes associated with the metabolic, plant–pathogen interaction, and hormone signaling pathways [12,13,38,51]. However, the function of DElncRNAs in RBSDV infection and the relationships of DElncRNA with DEmRNAs remain to be studied.

To identify the potential lncRNA-regulated mRNAs involved in RBSDV infection in rice, we conducted DElncRNA and mRNA co-location and co-expression analyses and mainly focused on DEmRNAs rather than all mRNAs. We found that plenty of DEmRNAs were co-expressed with specific DElncRNAs, while only a few were co-located with specific DElncRNAs (Figure 5, Table 3 and Appendix A), suggesting that the regulation of mRNA expression by lncRNAs is not only dependent on their chromosomal locations but also is affected by other unidentified and possibly complex co-expressed networks. In this study we also noticed that five mRNAs in the ubiquitin-mediated proteolysis (ko04120) were co-localized with DElncRNAs (Figure 4a and Appendix A), suggesting that RBSDV infection in rice can regulate several genes related to the ubiquitin–proteasome pathway through lncRNAs. To better understand these complex regulatory relationships of lncRNA to mRNA, the predicted co-expression network for the DElncRNAs and DEmRNAs was constructed (Figure 5 and Appendix A). Due to the large quantity of DElncRNA–DEmRNA co-expression pairs in this network, it is difficult for us to find the co-expression pairs of the DElncRNA–DEmRNA that are potentially related to RBSDV infection. To narrow the scope of genes in the co-expression network, we filtered out the DEmRNAs which were related to the plant–pathogen interaction (Pathway ID:ko004626) and constructed a new network for them (Figure 6 and Appendix A). It is noteworthy that most predicted DElncRNA–DEmRNA pairs were upregulated in this network, suggesting that RBSDV infection in rice could robustly activate many gene expressions. The LRR domain-containing proteins are important for protein recognition and play critical roles in plant defense. Some viral proteins, such as p50 of tobacco mosaic virus (TMV) and the coat protein of potato virus X (PVX), can be perceived by plant LRR proteins, and then the antiviral response will be activated [52]. We found that there were some LRR domain-containing proteins in this network, such as LOC_Os01g06730, LOC_Os08g10310, and LOC_Os02g17400 (Figure 6 and Appendix A). Thus, we speculate that these LRR domain-containing proteins might participate in the recognition of viral proteins. In our co-expression network, *OsCML16* (LOC_Os01g04330) and *OsCML21* (LOC_Os05g24780) encoding the plant-specific calmodulin-like proteins, were found to be co-expressed with four DElncRNAs (Appendix A). Calcium (Ca^2+^) is an important messenger involved in plant defense signal transduction, and some calmodulin-like proteins are necessary for the accumulation of reactive oxygen species (ROS) against pathogen invasion [53]. A previous study has shown that ROS is accumulated after RBSDV infection and enhances the resistance of rice to the virus [12]. These genes encoding the plant-specific calmodulin-like proteins may be upregulated by DElncRNAs to withstand viral infection by promoting ROS accumulation. We also found that genes participating in flavonoids biosynthesis could also be regulated by DElncRNAs, such as TCONS_00065705-LOC_Os03g18030, TCONS_00065705-LOC_Os10g16974, and TCONS_00059822-LOC_Os04g51660 (Figure 6 and Appendix A). The flavonoids are important in plant resistance against pathogens and insects [41], and have been shown to enhance the plant resistance to PVX and tomato spotted wilt virus (TSWV) [54,55], as well as the resistance of rice to the brown planthopper [56]. The upregulation of these DElncRNAs and their related flavonoid biosynthesis genes suggests that rice can increase the accumulation of the flavonoids to counter viral infection and planthopper feeding, and lncRNAs may be essential regulators in this process.

Besides the DElncRNAs participating in the antiviral response against RBSDV infection, we also found that some DElncRNAs may be utilized by RBSDV to enhance viral infection. Previous studies have indicated that plant hormones: ABA, JA, BR, and auxin, play important roles in RBSDV infection in rice plants [11,12,38]. In this study, we have found that some hormone signaling- and biosynthesis-related DEmRNAs were co-expressed with specific DElncRNAs, such as both *OsJAZ8* (LOC_Os09g26780) and *OsJAZ11* (LOC_Os03g08320) are co-expressed with two DElncRNAs (TCONS_00089479 and TCONS_00065705) (Figure 6 and Appendix A). The cytochrome P450 was reported to be implicated in brassinosteroids biosynthesis in rice [57], and in our results, a cytochrome *P450* (LOC_Os01g72270) co-located with TCONS_00011723 (Table 3). The JA signaling pathway has been proved to positively regulate antiviral defense against RBSDV in rice, while the BR signaling pathway was reported to mediate susceptibility to RBSDV infection [58]. Interestingly, JAZ protein is a repressor of the JA signaling pathway, and cytochrome P450 is involved in BR biosynthesis. The upregulation of these genes may inhibit the JA signaling pathway and promote the BR signaling pathway, which may contribute to RBSDV infection on rice. In another co-expressed lncRNA–mRNA pair, *OsWRKY28* (LOC_Os06g44010) was co-expressed with lncRNA TCONS_00065705 (Figure 6 and Appendix A). OsWRKY28 is reported to be a negative regulator of innate immune responses against rice blast fungus in rice [59], and thus we speculate that the upregulation of *OsWRKY28* would also enhance the susceptibility of rice to RBSDV.

Based on the computational prediction of DElncRNA–DEmRNA regulatory relationships, we attempted to confirm these relationships experimentally. Most studies identifying the lncRNAs in rice through high-throughput RNA-seq techniques did not verify whether the lncRNA–mRNA regulatory relationships truly exist or not, which is mainly because of the relatively long period of obtaining lncRNA-silenced transgenic rice plants. Here, we succeeded in silencing certain DElncRNAs in rice calli by *Agrobacterium*-mediated transformation, and then the expression levels of the predicted co-expressed DEmRNAs were detected through qRT-PCR. Three DElncRNAs were selected to be silenced, and eight co-expressed DEmRNAs were tested. As a result, we found that five DEmRNAs could be regulated by the selected DElncRNAs (Figure 8 and Table 4). Moreover, LOC_Os01g06730 predicted to encode a receptor-like protein was found to be regulated by two DElncRNAs (TCONS_00065705 and TCONS_00089479), suggesting the regulation of lncRNA to mRNA may be a complex relationship (Figure 6 and Figure 8, Table 4 and Appendix A). We also found that the DEmRNAs confirmed to co-express with DElncRNAs experimentally in this study were mainly located in the middle to the upstream of their related pathways, which indicates that lncRNAs are prone to alter pathways at their middle to upstream location. Genes downstream of a pathway may not be the target of lncRNAs. For example, the expression level of a *PR1* gene (LOC_Os01g28500) which locates at the downstream of the plant defense pathway, did not significantly change after silencing its predicted co-expressed DElncRNAs (TCONS_00065705 and TCONS_00089479) in our test (Figure 6 and Appendix A, Table 4 and Appendix A). Although not all selected co-expressed DEmRNAs were regulated by the corresponding DElncRNAs in our experiments, this method still verified a significant number of lncRNA–mRNA regulatory relationships. We think that our results here prove this method to be a feasible and efficient way to screen lncRNA–mRNA regulatory pairs for further study. Besides, this method is less time-consuming compared with the traditional transgenic method, which makes it possible to rapidly screen lncRNA–mRNA regulatory pairs from considerable computationally predicted data.

In brief, these results provide clues on the roles of DEmRNA and DElncRNA in confronting RBSDV infection and information for future functional studies of the candidate genes involved in rice-RBSDV interaction. Moreover, we also present a new method to rapidly screen lncRNA–mRNA regulatory pairs based on transient gene silencing in rice calli through *Agrobacterium*-mediated transformation, which will be helpful to the function study of lncRNAs in rice in the future. To understand the interaction between rice and RBSDV, further investigations on regulatory mechanisms of lncRNA–mRNA and the functions of their targeted pathways are necessary.

## Figures and Tables

**Figure 1 viruses-12-00951-f001:**
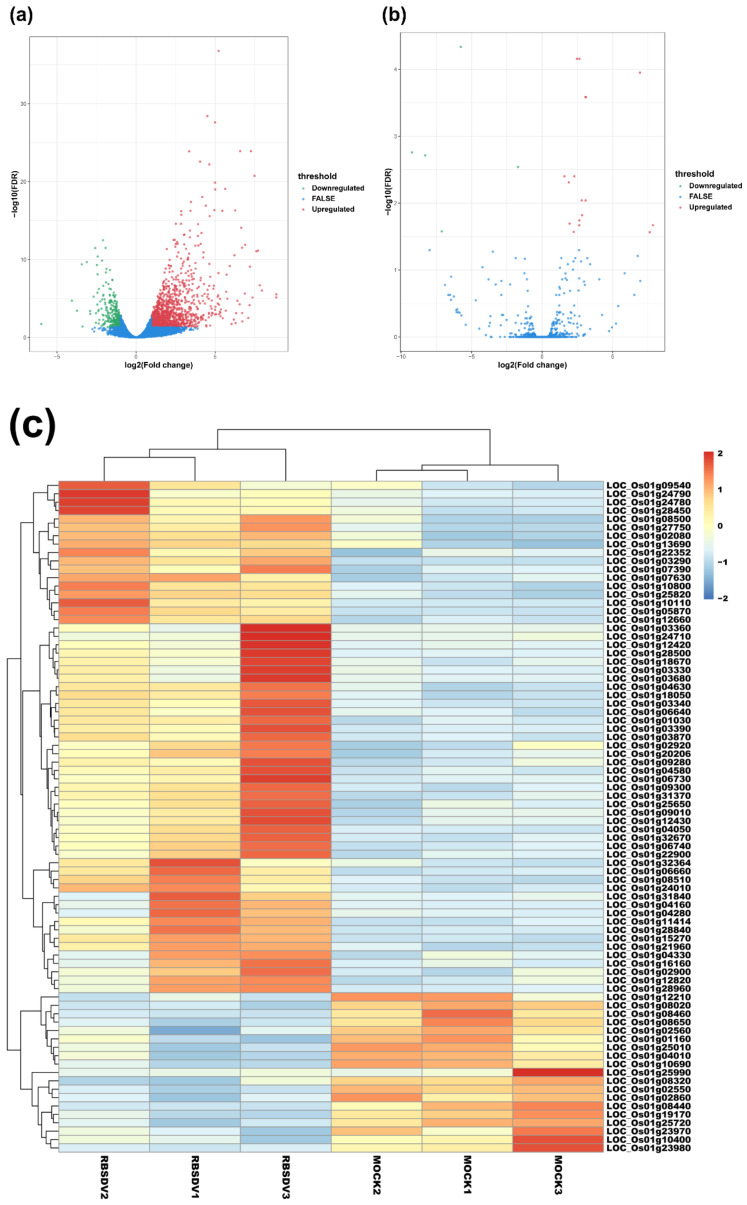
Volcano maps and hierarchical clustering analysis show expression profiles of mRNAs and lncRNAs of RBSDV-infected and mock rice plants. Volcano maps show mRNA expression (**a**) and lncRNA expression (**b**) with fold change ≥ 2.0 and false discovery rate (FDR) < 0.05. Red dots represent upregulated genes, and green dots represent downregulated genes in volcano maps. The heatmaps show significantly expressed mRNAs (**c**) and lncRNAs (**d**) with fold change ≥ 2.0 and FDR < 0.05. The right panel is the color key for fold-change. Red represents upregulation, blue represents downregulation, the brighter image is the greater the fold change; (**e**,**f**) the number of mRNAs or lncRNAs whose expression significantly changed are shown.

**Figure 2 viruses-12-00951-f002:**
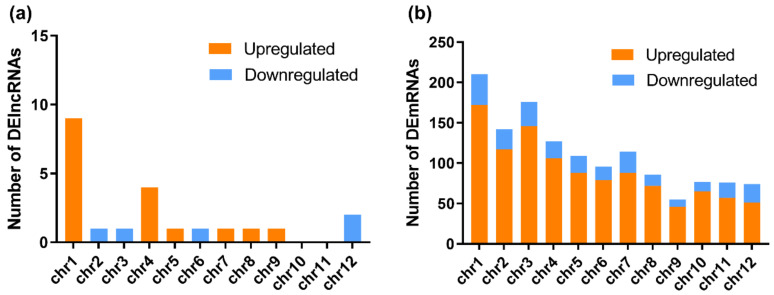
Chromosomal distributions of DElncRNAs and differentially expressed mRNAs (DEmRNAs). The numbers of DElncRNAs (**a**) and DEmRNAs (**b**) on each chromosome are shown. Orange represents upregulation, while blue represents downregulation.

**Figure 3 viruses-12-00951-f003:**
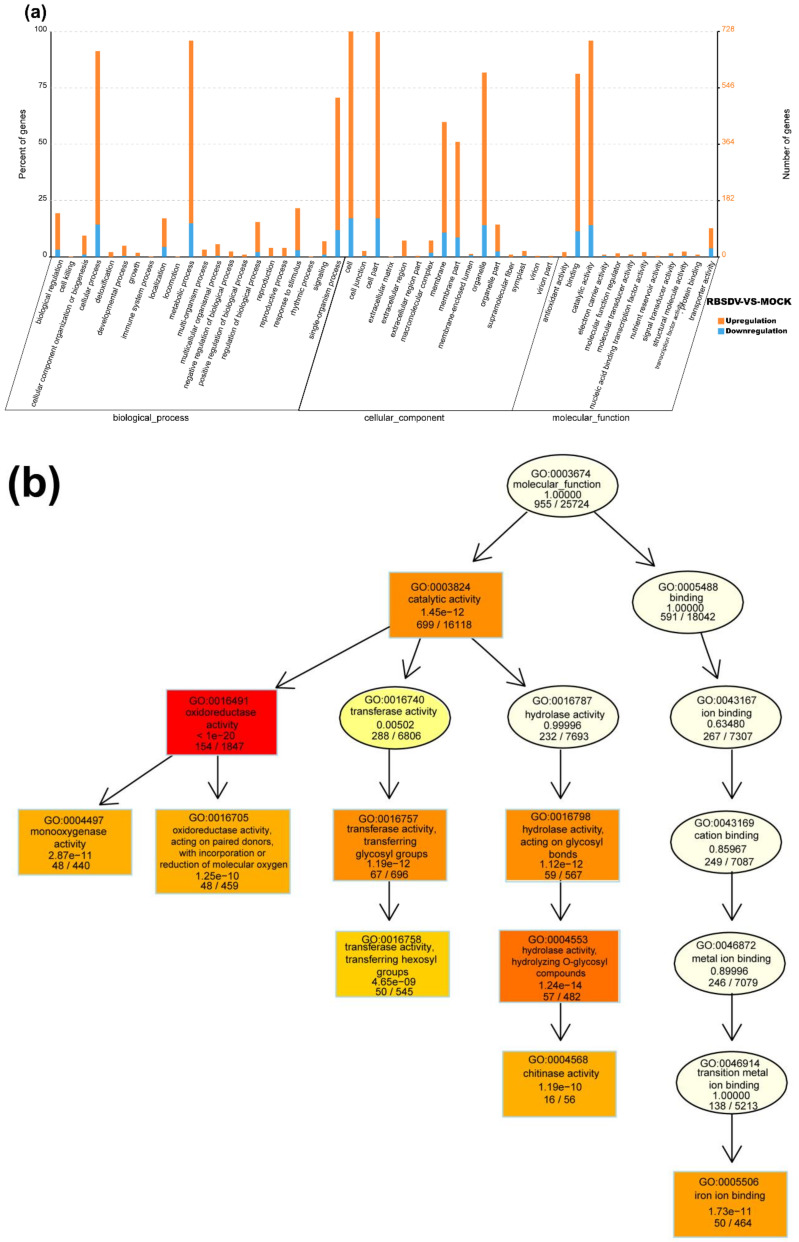
Gene ontology (GO) function enrichment analysis, GO terms, and KEGG pathway enrichment of DEmRNAs identified in RBSDV-infected rice plants. (**a**) GO enrichment analysis of DEmRNAs with a *p*-value < 0.05; (**b**) gene ontology terms for molecular function of DEmRNAs; (**c**) statistics of the top 20 KEGG pathways enriched for DEmRNAs. The size of each circle represents the number of significant DEmRNAs enriched in the corresponding pathway. The enrichment factor was calculated using the number of enriched DEmRNAs divided by the total number of background genes in the corresponding pathway. A pathway with a *p*-value < 0.05 is considered to be significantly over-represented.

**Figure 4 viruses-12-00951-f004:**
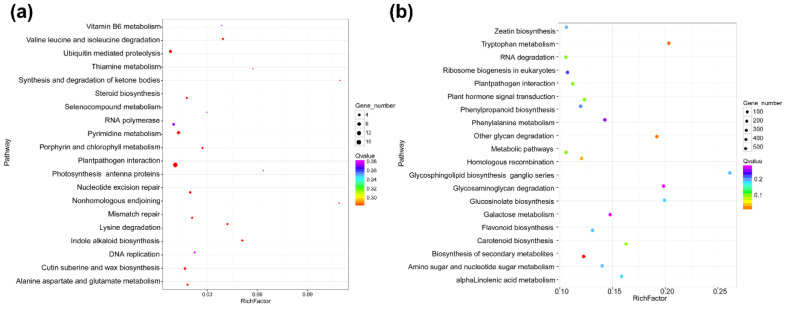
Statistics of the KEGG pathways enriched for mRNAs co-located with DElncRNAs (**a**) and mRNAs co-expressed with DElncRNAs (**b**). The size of each circle represents the number of significant DEmRNAs enriched in the corresponding pathway. The enrichment factor was calculated using the number of enriched DEmRNAs divided by the total number of background genes in the corresponding pathway. A pathway with a *p*-value < 0.05 is considered to be significantly over-represented.

**Figure 5 viruses-12-00951-f005:**
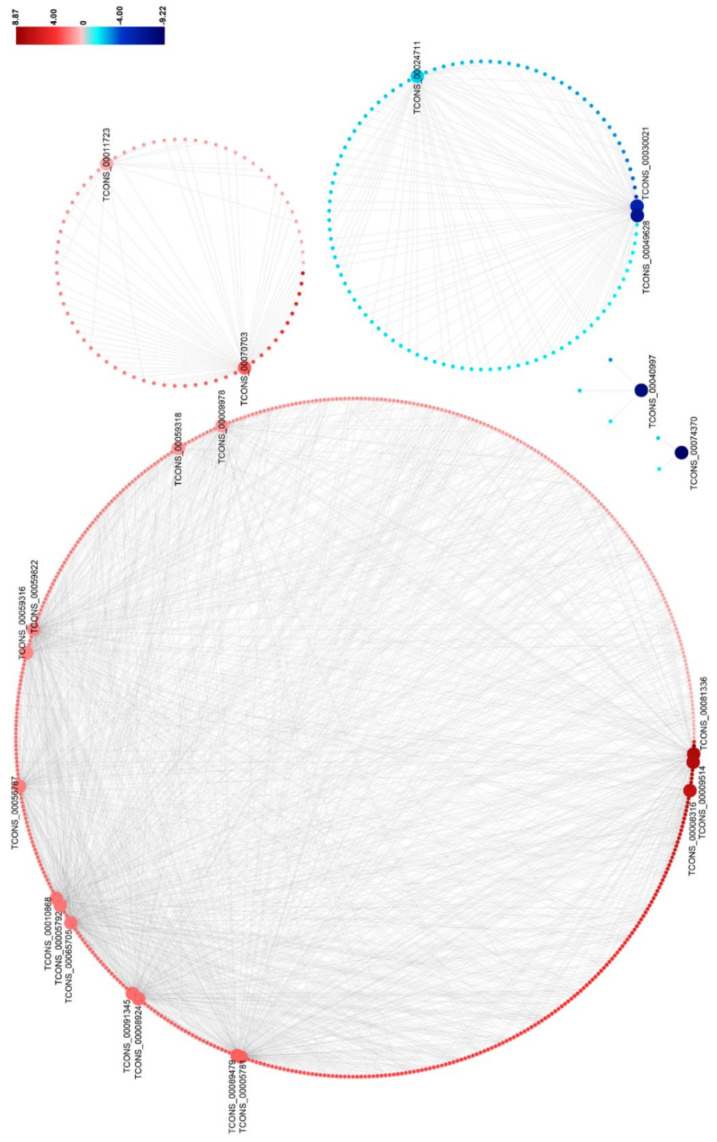
The network for the predicted co-expressed DElncRNA–DEmRNA pairs found in RBSDV-infected rice plants. A fold-change scale is shown on the right side. Red indicates upregulation and blue indicates downregulation. The brighter image is the greater fold change. A DElncRNA–DEmRNA pair with a person’s correlation coefficient > 0.95 or < −0.95 is considered to be co-expressed.

**Figure 6 viruses-12-00951-f006:**
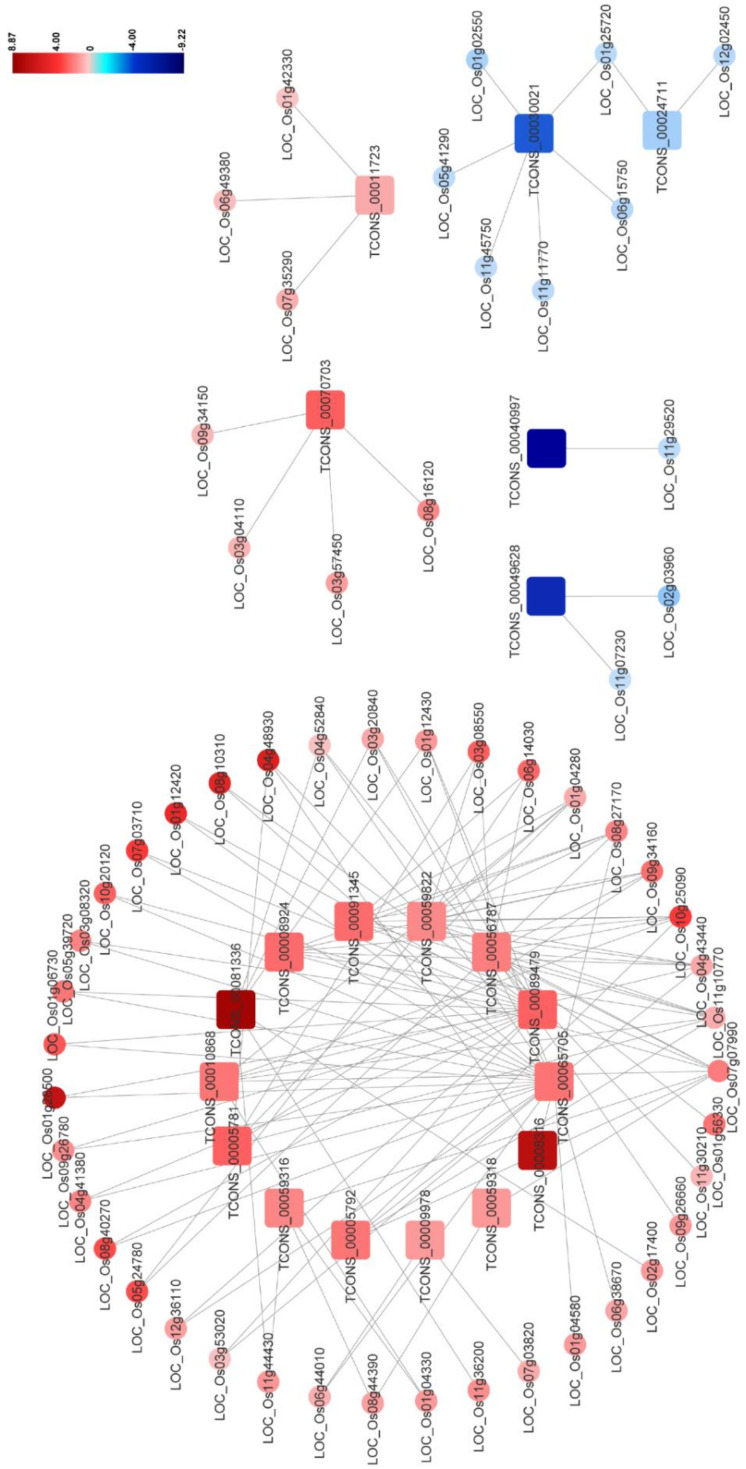
The network for the predicted DElncRNA–DEmRNA co-expression pairs potentially involved in plant–pathogen interaction (Pathway ID:ko004626). The square represents DElncRNA and the circle represents DEmRNA. Red indicates upregulation and blue indicates downregulation. The brighter image is the greater fold change. Each selected pair is with a person’s correlation coefficient > 0.95 or < −0.95. Identitiess of DElncRNAs and DEmRNAs are shown in the figure.

**Figure 7 viruses-12-00951-f007:**
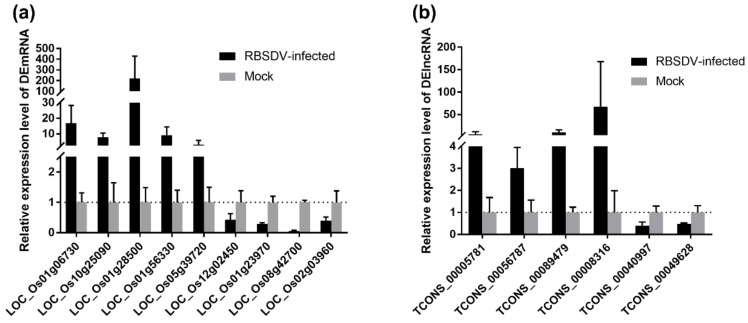
Validation of expression of selected DEmRNAs (**a**) and DElncRNAs (**b**) though qRT-PCR. The total RNA from RBSDV-infected and mock rice plants was extracted for qRT-PCR analyses at 30 dpi. *OsUBC* gene served as an internal reference for the relative quantification. The values represented means of the gene expression levels ± standard deviations (SD) relative to the mock plants (*n* = 3 biological replicates).

**Figure 8 viruses-12-00951-f008:**
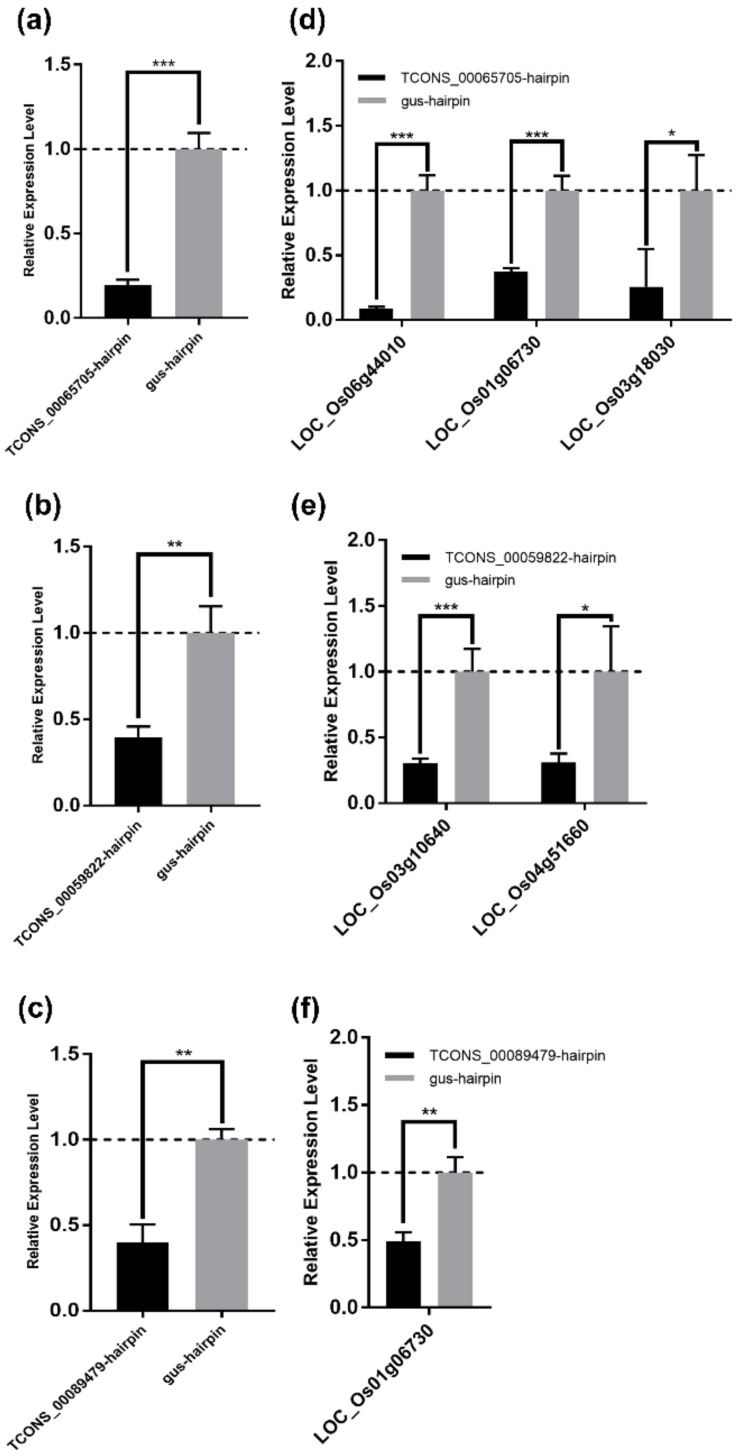
Experimental validation of the lncRNA–mRNA regulatory relationship in the DElncRNA–DEmRNA co-expression network. (**a**–**c**) The silencing of the lncRNA TCONS_00065705 (**a**), TCONS_00059822 (**b**), and TCONS_00089479 (**c**) was confirmed by qRT-PCR. Total RNA of rice callus was extracted at 7 days after *Agrobacterium* infection. *OsUBC* gene served as an internal reference for the relative quantification. The values represent means of the lncRNA expression levels ± standard deviations (SD) relative to the gus-hairpin-expressed callus tissues (*n* = 3 biological replicates). Data were analyzed by the Student’s *t*-test, and asterisks denote significant differences between lncRNA-silenced and gus-hairpin-expressed callus tissues (two-sided, ** *p* < 0.01, *** *p* < 0.001). (**d**–**f**) qRT-PCR analyses of the mRNA co-expressed with TCONS_00065705 (**d**); TCONS_00059822 (**e**); and TCONS_00089479 (**f**). Total RNA in (**a**–**c**) was used for qRT-PCR analyses. *OsUBC* gene served as an internal reference for the relative quantification. The values represent means of the mRNA expression levels ± standard deviations (SD) relative to the gus-hairpin-expressed callus tissues (*n* = 3 biological replicates). Data were analyzed by the Student’s *t*-test, and asterisks denote significant differences between lncRNA-silenced and gus-hairpin-expressed callus tissues. (two-sided, * *p* < 0.05, ** *p* < 0.01, *** *p* < 0.001).

**Table 1 viruses-12-00951-t001:** Summary of RNA sequencing results.

	Mock-Inoculated			RBSDV-Inoculated		
	MOCK1	MOCK2	MOCK3	RBSDV1	RBSDV2	RBSDV3
Total Reads	52,107,508	50,199,476	50,199,546	48,580,152	50,198,952	48,580,180
Clean Reads	47,629,488	46,285,766	46,451,138	45,328,332	46,432,674	45,452,636
Reads-rRNAs	46,936,588	45,549,054	45,510,752	44,740,026	45,793,276	44,324,792
Reads Mapped to Rice Genome	42,336,802 (90.20%)	37,350,224 (82.00%)	36,545,133 (80.30%)	36,060,460 (80.60%)	33,291,711 (72.70%)	29,520,311 (66.60%)
Q20 (%)	97.37	96.28	96.25	96.48	96.15	96.42
Q30 (%)	92.65	90.14	90.10	90.62	89.85	90.49

Note: Total Reads, raw data from the Illumina sequencing; Clean Reads, reads obtained after quality control filtering; Reads-rRNAs, reads obtained after removing ribosomal RNA; Q20(%), clean reads with 99% accuracy; Q30(%), clean reads with 99.9% accuracy.

**Table 2 viruses-12-00951-t002:** Identified differentially expressed long non-coding RNAs (lncRNAs) (DElncRNAs) in the Rice black-streaked dwarf virus (RBSDV)-infected rice plants.

Gene ID	Length	log2Ratio (RBSDV/MOCK)	Up-Down-Regulation (RBSDV/MOCK)	*p*-Value	FDR	Chromosome Location
TCONS_00081336	4448	7.86	up	3.21 × 10^−4^	2.14 × 10^−2^	chr7
TCONS_00009514	714	7.62	up	4.88 × 10^−4^	2.71 × 10^−2^	chr1
TCONS_00008316	1887	6.93	up	3.67 × 10^−7^	1.12 × 10^−4^	chr1
TCONS_00070703	1608	3.10	up	1.27 × 10^−6^	2.60 × 10^−4^	chr8
TCONS_00005781	11439	3.08	up	1.04 × 10^−4^	9.10 × 10^−3^	chr1
TCONS_00089479	1652	3.07	up	1.27× 10^−6^	2.60 × 10^−4^	chr1
TCONS_00008924	1722	2.83	up	1.86 × 10^−4^	1.52 × 10^−2^	chr1
TCONS_00091345	4501	2.81	up	9.69 × 10^−5^	9.10 × 10^−3^	chr9
TCONS_00065705	430	2.65	up	1.71 × 10^−7^	6.96 × 10^−5^	chr5
TCONS_00005792	12346	2.64	up	2.36 × 10^−4^	1.81 × 10^−2^	chr1
TCONS_00010868	1028	2.63	up	3.32 × 10^−4^	2.14 × 10^−2^	chr1
TCONS_00056787	4596	2.48	up	1.30 × 10^−7^	6.96 × 10^−5^	chr4
TCONS_00059316	5727	2.28	up	3.32 × 10^−5^	3.97 × 10^−3^	chr4
TCONS_00059822	2929	2.24	up	4.63 × 10^−4^	2.69 × 10^−2^	chr4
TCONS_00059318	11526	1.95	up	2.80 × 10^−4^	2.02 × 10^−2^	chr4
TCONS_00009978	1177	1.89	up	4.79 × 10^−5^	4.89 × 10^−3^	chr1
TCONS_00011723	2453	1.58	up	3.57 × 10^−5^	3.97 × 10^−3^	chr1
TCONS_00024711	4310	−1.71	down	2.11 × 10^−5^	2.87 × 10^−3^	chr12
TCONS_00030021	3026	−5.76	down	3.77 × 10^−8^	4.61 × 10^−5^	chr12
TCONS_00049628	1917	−7.12	down	4.32 × 10^−4^	2.64 × 10^−2^	chr3
TCONS_00040997	10059	−8.28	down	1.26 × 10^−5^	1.93 × 10^−3^	chr2
TCONS_00074370	6771	−9.22	down	9.99 × 10^−6^	1.75 × 10^−3^	chr6

**Table 3 viruses-12-00951-t003:** The co-located DElncRNA–DEmRNA pairs.

DElncRNA				Co-Located DEmRNA				
lncRNA ID	lncRNA Exon Position	Up or Down-Regulation (RBSDV/MOCK)	log2Ratio (RBSDV/MOCK)	mRNA_ID	mRNA_Position	Up or Down-Regulation (RBSDV/MOCK)	log2Ratio (RBSDV/MOCK)	Gene Product Name
TCONS_00089479	17556112–17556165, 17558595–17560192	up	3.07	LOC_Os08g28710	17558062–17560192	up	3.06	Receptor protein kinase CRINKLY4 precursor
TCONS_00065705	8902479–8902875, 8949120–8949152	up	2.65	LOC_Os05g15770	8902479–8903793	up	2.31	Glycosyl hydrolase
TCONS_00010868	36308049–36308470, 36310641–36311246	up	2.63	LOC_Os01g62630	36268063–36269737	up	2.97	Aspartic proteinase nepenthesin precursor
TCONS_00056787	6938662–6942391, 6942487–6943054, 6943635–6943932	up	2.48	LOC_Os04g12540	6939046–6942716	up	2.43	Receptor-like protein kinase
TCONS_00059316	27164826–27165925, 27166037–27166095, 27166309–27166512, 27171990–27175910, 27177026–27177468	up	2.28	LOC_Os04g45890	27172115–27173657	up	1.73	Retrotransposon protein
TCONS_00059822	30607493–30609094, 30609233–30610559	up	2.24	LOC_Os04g51660	30608765–30610406	up	1.79	Transferase family protein
TCONS_00059822	30607493–30609094, 30609233–30610559	up	2.24	LOC_Os04g51796	30697527–30701645	up	6.33	DNA repair ATPase-related
TCONS_00009978	30512276–30512751, 30512877–30513308, 30513415–30513683	up	1.89	LOC_Os01g53090	30512276–30521565	up	1.59	Pathogen-related protein
TCONS_00011723	41949333–41951662, 41951830–41951887, 41952060–41952124	up	1.58	LOC_Os01g72270	41900002–41901651	up	3.89	Cytochrome P450-dependent fatty acid hydroxylase
TCONS_00024711	819049–819195, 819546–819756, 820494–824445	down	−1.71	LOC_Os12g02330	746155–747300	up	1.47	LTPL13—Protease inhibitor
TCONS_00030021	19637837–19638788, 19638882–19640733, 19641766–19641987	down	−5.76	LOC_Os12g32610	19675353–19676314	up	2.57	-
TCONS_00049628	22799469–22800028, 22800496–22801659, 22803529–22803721	down	−7.12	LOC_Os03g40930	22744328–22746916	up	1.49	-

**Table 4 viruses-12-00951-t004:** List of tested DElncRNAs and DEmRNAs.

DElncRNA	DEmRNA	Verified?	Gene Product Name
TCONS_00065705	LOC_Os06g44010	Yes	OsWRKY28
	LOC_Os01g06730	Yes	Receptor-like protein
	LOC_Os03g18030	Yes	Leucocyanidin oxygenase
	LOC_Os01g28500	No	PR1 protein
	LOC_Os03g03034	No	Flavonol synthase/flavanone 3-hydroxylase
TCONS_00059822	LOC_Os03g10640	Yes	Calcium-transporting ATPase
	LOC_Os04g51660	Yes	Malonyl transferase
	LOC_Os02g47470	No	Cytochrome P450
TCONS_00089479	LOC_Os01g06730	Yes	Receptor-like protein
	LOC_Os01g28500	No	PR1 protein

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
