# Peer review of "Transcriptome Analysis of Rice Reveals the lncRNA–mRNA Regulatory Network in Response to Rice Black-Streaked Dwarf Virus Infection"

_viruses, 2020, doi:10.3390/v12090951_

Round 1
Reviewer 1 Report
This study used RNA-seq to identify DEmRNAs and DElncRNAs during RBSDV infection. The authors developed a method to identify mRNAs controlled by lncRNA expression. This study relied heavily on GO Term analysis, KEGG, etc. to draw conclusions on the significance of the RNA-seq results. The data that is presented is overwhelming in that heatmaps, gene networks, etc contain up to thousands of entries but fail to draw specific conclusions. The manuscript could be greatly improved by focusing on specific results (i.e. the DEmRNA-DElncRNA co-expression) versus presenting expansive GO term analyses, heatmaps, etc.
Lines 56-57: re-word. Should read “Recent transcriptome-wide studies using high-throughput RNA-seq have proved that the plant genome transcribes many lncRNAs”
Fig. 1A-B: The figure legend must state what cutoffs were used in the volcano plots that correspond with the coloring.
Fig. 1C-D: This figure presents multiple issues. First, the text is too small to read. Second, the data presented serves little purpose other than showing that many genes are up- or down-regulated during RBSDV infection. Based on all earlier studies using RNA-seq with RNA virus infections, this is expected. This figure should overlay up- or down-regulated pathways with the data presented to provide useful information to the reader. If not, a Principal Component Analysis (PCA) plot of the RNA-seq samples would show the degree of correlation within and between groups and would be much easier to interpret.
Fig. 1F: The number of DElncRNAs identified is drastically different compared to the lncRNA study in Arabidopsis cited earlier in the text (Line 64). Based on the data presented, nearly 100-fold fewer DElncRNAs were identified during RBSDV infection compared to various stress treatments in Arabidopsis like cold, high-salt, and abscisic acid treatment. This reviewer would expect a significantly higher number of DElncRNAs. How reliable is your method for identifying lncRNAs? Is it possible that your cutoffs were too stringent when identifying lncRNAs?
Lines: 258-263/ Fig. 2: Pertaining to my points above, the distribution of lncRNAs across chromosomes lacks statistical power as the sample size (n=22) is very low. Are these distributions significantly different compared to random distributions?
Fig. 3: Again, the text is small (especially Fig. 3B). The data presented seemingly suggests that RBSDV affects multiple pathways and processes, but the authors fail to note which pathways are known to be significant for virus infection, other than chitinase. Also, without P values in Fig.3A, it is not possible to assess whether the DEmRNAs are actually enriched in these processes, or functions. Fig. 3C again seems to show a wide-range of processes affected by RBSDV, but the authors do not attempt correlate these results with the biology of RBSDV.
Lines 317-343: This is an interesting approach to linking gene expression with the DElncRNAs. The authors did correlate the putative genes controlled by DElncRNAs with RBSDV infection (e.g. P450, flavonoid biosynthesis). However, the interesting findings are lost in Fig. 4. Fig. 4 presents a large number of possible pathways affected by mRNAs co-located with DElncRNAs, but fails to draw specific conclusions.
Lines 353-361: It is unclear what the PCC demonstrates. How were co-expressed pairs identified? Does this mean they had similar chromosomal locations and similar changes in gene expression? This should be clarified.
Figs. 5-6: The data presented is extensive and could be simplified as “lncRNAs possibly regulate hundreds of mRNAs”. The large networks are almost entirely un-validated, and this reviewer finds the results presented to be primarily speculative.
Lines 436-455: It would be interesting to know the effects of lncRNA-silencing on RBSDV infection. Does silencing the DElncRNA-DEmRNA co-expression reduce or improve virus fitness?
Lines 578-583: This reviewer questions the efficiency and feasibility of using this method to test many pairs of DElncRNAs and DEmRNAs. This method requires cloning hairpin constructs, agrobacterium-transformation, followed by RT-qPCR. The cloning of hairpin constructs alone would be a significant bottleneck for testing many pairs.
Author Response
Reviewer: #1
Comments and Suggestions for Authors
This study used RNA-seq to identify DEmRNAs and DElncRNAs during RBSDV infection. The authors developed a method to identify mRNAs controlled by lncRNA expression. This study relied heavily on GO Term analysis, KEGG, etc. to draw conclusions on the significance of the RNA-seq results. The data that is presented is overwhelming in that heatmaps, gene networks, etc contain up to thousands of entries but fail to draw specific conclusions. The manuscript could be greatly improved by focusing on specific results (i.e. the DEmRNA-DElncRNA co-expression) versus presenting expansive GO term analyses, heatmaps, etc.
OUR RESPONSE: We would like to thank the respected reviewer #1 for your helpful comments. We presented the detailed information of transcriptome analyses, such as. heatmaps, gene networks, etc. Because we think this is not only important to the completeness of the transcriptomic analysis, but is also useful to the reader who is intended to mine additional information from this study. We thank the respected reviewer #1 for your positive comments.
The respected reviewer think that we fail to draw specific conclusions. This issue may be a common problem in transcriptomic studies. But, we tried to solve it by narrowing the range of potential genes step by step. We first identified 22 DElncRNAs and 1342 DEmRNAs, and constructed a DElncRNA-DEmRNA co-expression network in RBSDV-infected rice plants. Because there are too many genes in this DElncRNA-DEmRNA co-expression network, we then narrowed the range of the network by screening out 56 DEmRNAs related to the plant-pathogen interaction pathway, and constructed a network for the DElncRNA–DEmRNA co-expression pairs potentially involved in plant-pathogen interaction. It is difficult to determine whether a gene is important in a certain process just by bioinformatics analysis, and appropriate biological experiments is needed. In this study, we screened some lncRNA-mRNA pairs which might regulate the response of rice to RBSDV infection, and the regulatory relationships of six DElncRNA–DEmRNA pairs were validated, such as TCONS_00065705-LOC_Os01g06730, TCONS_00065705-LOC_Os06g44010, and TCONS_00089479- LOC_Os01g06730. We think it is very likely that these lncRNA-mRNA pairs might participate in rice-RBSDV interactions, but extra biological experiments (i.e. challenging gene knockout rice with RBSDV) will be done in our further study.
Lines 56-57: re-word. Should read “Recent transcriptome-wide studies using high-throughput RNA-seq have proved that the plant genome transcribes many lncRNAs”
OUR RESPONSE: We have revised accordingly.
Fig. 1A-B: The figure legend must state what cutoffs were used in the volcano plots that correspond with the coloring.
OUR RESPONSE: We have added the cutoffs i.e. fold change ≥ 2.0 and FDR < 0.05 in the figure legend.
Fig. 1C-D: This figure presents multiple issues. First, the text is too small to read. Second, the data presented serves little purpose other than showing that many genes are up- or down-regulated during RBSDV infection. Based on all earlier studies using RNA-seq with RNA virus infections, this is expected. This figure should overlay up- or down-regulated pathways with the data presented to provide useful information to the reader. If not, a Principal Component Analysis (PCA) plot of the RNA-seq samples would show the degree of correlation within and between groups and would be much easier to interpret.
OUR RESPONSE: The figure has been improved. In figure 1C and 1D, we present the heatmaps as the visual representation of the differentially expressed transcripts in RBSDV-infected rice plants, which is a commonly used method in most transcriptomic studies. Certainly, many upregulation and downregulation of mRNAs in RBSDV-infected rice plants are consistent with the results from plants infected with other different viruses. But, there are some mRNAs with different expression patterns in RBSDV-infected rice plants. Thus, the heatmap of plant transcripts after different virus infection is not exactly same. Moreover, we present the heatmap of rice lncRNAs after RBSDV infection, which is not ever showed in other reported studies. Therefore, this figure is useful to the researcher who is intended to mine additional information for his/her study. In addition, two Principal Component Analysis (PCA) figures of the RNA-seq samples have been added in the supplementary data.
Fig. 1F: The number of DElncRNAs identified is drastically different compared to the lncRNA study in Arabidopsis cited earlier in the text (Line 64). Based on the data presented, nearly 100-fold fewer DElncRNAs were identified during RBSDV infection compared to various stress treatments in Arabidopsis like cold, high-salt, and abscisic acid treatment. This reviewer would expect a significantly higher number of DElncRNAs. How reliable is your method for identifying lncRNAs? Is it possible that your cutoffs were too stringent when identifying lncRNAs?
OUR RESPONSE: We used a method with the common threshold of |log2(fold change)| ≥ 1 and a statistical significance (FDR < 0.05) to identify lncRNAs. We know that 1832 is a total number of altered lncRNA in Arabidopsis thaliana under different treatments (drought, cold, high-salinity and ABA treatments) and different time (2h/10h) (Plant Cell 2012, 24, 4333–4345). In another related paper published in 2008, the number of differentially altered lncRNA is not similar under different treatments (Plant Cell Physiol. 2008, 49, 1135–1149). For example, there were 479 upregulated and 78 downregulated lncRNAs under the drought stress, while only 44 upregulated and 9 downregulated lncRNAs were found under the cold stress (Plant Cell Physiol. 2008, 49, 1135–1149). Thus, we think it is very possible that there are relatively few DElncRNAs in response to RBSDV infection.
Lines: 258-263/ Fig. 2: Pertaining to my points above, the distribution of lncRNAs across chromosomes lacks statistical power as the sample size (n=22) is very low. Are these distributions significantly different compared to random distributions?
OUR RESPONSE: Among these identified 22 DElncRNAs, the 17 upregulated DElncRNAs were found on chromosomes 1, 4, 5, 7-9, while the 5 downregulated DElncRNAs were distributed on chromosomes 2, 3, 6, and 12 (Figure 2a, Table 2). However, no DElncRNA is distributed on chromosomes 10 and 11, and no rice chromosome has upregulated and downregulated DElncRNAs simultaneously. Thus, we think that these distributions are significantly different compared to random distributions.
Fig. 3: Again, the text is small (especially Fig. 3B). The data presented seemingly suggests that RBSDV affects multiple pathways and processes, but the authors fail to note which pathways are known to be significant for virus infection, other than chitinase. Also, without P values in Fig.3A, it is not possible to assess whether the DEmRNAs are actually enriched in these processes, or functions. Fig. 3C again seems to show a wide-range of processes affected by RBSDV, but the authors do not attempt correlate these results with the biology of RBSDV.
OUR RESPONSE: The figure has been improved, and the p-value is added in Figure 3A legend. The Go and KEGG enrichment analyses of DEmRNAs are the necessary approaches to discover regulated cell processes and pathways in RBSDV-infected rice plants. Using this figure, we would like to give an overall description about identified DEmRNAs in RBSDV-infected rice plants. In the revised manuscript, we have added the content about some pathways which are significant for RBSDV infection in our manuscript: Reactive oxygen species (ROS) plays multitude roles in rice defense response to RBSDV infection. A lot of DEmRNAs were classified into the oxidoreductase activity (GO: 0016491), suggesting the regulation of cell redox status in response to RBSDV infection (Fig. 3a, b). Previous studies demonstrated that plant hormones played important roles in plant defense against RBSDV infection. Jasmonic acid and auxin signaling pathways enhance the resistance of rice to RBSDV, while abscisic acid and brassinosteroid pathways mediate the susceptibility to RBSDV infection. In our results, we found that the DEmRNAs were also enriched in plant hormone signal transduction pathway (ko04075), indicating broad responses of plant hormone signaling pathways to RBSDV infection (Fig. 3c).
Lines 317-343: This is an interesting approach to linking gene expression with the DElncRNAs. The authors did correlate the putative genes controlled by DElncRNAs with RBSDV infection (e.g. P450, flavonoid biosynthesis). However, the interesting findings are lost in Fig. 4. Fig. 4 presents a large number of possible pathways affected by mRNAs co-located with DElncRNAs, but fails to draw specific conclusions.
OUR RESPONSE: The mRNAs regulated by DElncRNAs are investigated in two aspects: one is their co-location and the other is their co-expression. Figure 4a shows top 20 enriched pathways of mRNAs co-located with DElncRNAs, and the co-located mRNAs involved in the plant-pathogen interaction and the ubiquitin-proteasome protein degradation pathway were more enriched compared with other pathways. Many studies have shown that the ubiquitin-proteasome protein degradation pathway plays an important role in the degradation of viral proteins. In contrast, viruses can also hijack plant ubiquitin-proteasome machinery to help their infections. Recently, He and colleagues reported that the P5-1 protein encoded by RBSDV could regulate the ubiquitination activity of SCF E3 ligase to inhibit the jasmonate signaling pathway to promote RBSDV infection. In this study we also noticed that five mRNAs in the ubiquitin-mediated proteolysis (ko04120) were co-localized with DElncRNAs (Figure 4a, Table S7), suggesting that RBSDV infection in rice can regulate several genes related to the ubiquitin-proteasome pathway through lncRNAs. Figure 4b shows top 20 enriched pathways of mRNAs co-expressed with DElncRNAs, and the DElncRNA–DEmRNA co-expression network is analyzed based on these enriched pathways in Figure 4b.
Lines 353-361: It is unclear what the PCC demonstrates. How were co-expressed pairs identified? Does this mean they had similar chromosomal locations and similar changes in gene expression? This should be clarified.
OUR RESPONSE: Pearson’s correlation coefficient (PCC) refers to the correlation coefficient between the expression levels of DElncRNA and mRNA data (New Phytol. 2015, 207, 1181-1197). In this analysis, the lncRNA-mRNA PCC was calculated using six lncRNA and six mRNA data respectively from six sequenced rice plants, and the lncRNA-mRNA pair with a PCC>0.95 or <-0.95 was defined as a co-expressed pair. The PCC of each lncRNA-mRNA pair is related to its expression levels in RBSDV-infected and mock plants and is independent of chromosomal locations.
Figs. 5-6: The data presented is extensive and could be simplified as “lncRNAs possibly regulate hundreds of mRNAs”. The large networks are almost entirely un-validated, and this reviewer finds the results presented to be primarily speculative.
OUR RESPONSE: In figures 5 and 6, a large number of lncRNA-mRNA co-expression pairs are predicted by bioinformatics, and here we hope to provide the complete information to readers who interest in rice-RBSDV interaction, vector viral transmission, etc. As we mainly interested in the plant-pathogen interaction pathway, and in the following research of this manuscript, we picked out some lncRNAs and mRNAs related to this pathway for further validation using a new approach based on transient gene silencing in rice calli. Our results indicate that lncRNA-mRNA regulatory relationships can be confirmed by combining bioinformatics and biological methods.
Lines 436-455: It would be interesting to know the effects of lncRNA-silencing on RBSDV infection. Does silencing the DElncRNA-DEmRNA co-expression reduce or improve virus fitness?
OUR RESPONSE: We would like to thank the respected reviewer #1 for your reasonable and useful comments. We suspect that it is very likely that silencing the DElncRNA-DEmRNA co-expression may reduce or improve virus fitness. Therefore, this is our following study.
Lines 578-583: This reviewer questions the efficiency and feasibility of using this method to test many pairs of DElncRNAs and DEmRNAs. This method requires cloning hairpin constructs, agrobacterium-transformation, followed by RT-qPCR. The cloning of hairpin constructs alone would be a significant bottleneck for testing many pairs.
OUR RESPONSE: It takes less than a week to construct hairpin constructs using the in-fusion cloning method, and whole experimental process for testing many pairs of DElncRNAs and DEmRNAs can be finished within 20 days, which is much faster than the conventional transgenic method as you know. Thus, we think that our method for testing the expression level of DElncRNA and DEmRNA pair is efficient and feasible.
Reviewer 2 Report
This interesting paper describes analysis of small RNA expression and regulation in rice infected with RBSDV. The following comments are meant to improve the manuscript:
- English language requires some small improvement. For example, the first sentence in the abstract, last word should be expression, not expressions.
- How would you anticipate RNA levels changing with a resistant variety of rice? Or if you constructed a transgenic plant with portions of the RBSDV genome? Would you be able to identify which part of the genome is responsible for changes in RNA regulation?
- Looking at regulatory pairs via computational analysis is an interesting approach. Can this be used to alter plants to have improved agronomic and nutritional benefits?
- some more information regarding the vertical axes in Figure 7 (Relative Expression) is needed.
Author Response
Reviewer: #2
Comments and Suggestions for Authors
This interesting paper describes analysis of small RNA expression and regulation in rice infected with RBSDV. The following comments are meant to improve the manuscript:
1. English language requires some small improvement. For example, the first sentence in the abstract, last word should be expression, not expressions.
OUR RESPONSE: We sincerely thank the respected reviewer #2 for taking his/her valuable time to point out the mistakes in our manuscript. During this revision, we tried very hard on the language of the manuscript. We have also obtained useful help from our colleagues and a native English speaking friend. We hope this revision meets the journal requirement.
2. How would you anticipate RNA levels changing with a resistant variety of rice? Or if you constructed a transgenic plant with portions of the RBSDV genome? Would you be able to identify which part of the genome is responsible for changes in RNA regulation?
OUR RESPONSE: To our knowledge, so far there are no anti-RBSDV resistant rice varieties reported. Thus, we can’t perform the transcriptome analyses of anti-RBSDV resistant rice varieties in this study. It is possible that the plant would upregulate some resistance genes by altering their co-expressed lncRNAs in rice plants. We also think that it is worthwhile to compare the expression levels of defense-related lncRNA-mRNA pairs between resistant and susceptible varieties if there is a resistant rice variety, which will be helpful to illuminate the defense mechanism of rice against RBSDV infection and anti-RBSDV resistant rice breeding. Certainly, the regulation of lncRNAs and mRNAs in rice is affirmatively caused by RBSDV-encoded proteins or its genome. The construction of transgenic plants is a practicable way to identify which part of the genome is responsible for changes in RNA regulation in RBSDV-infected rice plants. Our further work focuses on this scientific problem posed by the reviewer here.
3. Looking at regulatory pairs via computational analysis is an interesting approach. Can this be used to alter plants to have improved agronomic and nutritional benefits?
OUR RESPONSE: We would like to thank the respected reviewer #2 for your positive comment. Computational analysis followed by experimental verification is proved to be a feasible way to screen lncRNA-mRNA regulatory pairs of interest in this study. Thus, we believe that the approach developed in our study can be used to improve the agronomic and nutritional benefits of crops.
4. some more information regarding the vertical axes in Figure 7 (Relative Expression) is needed.
OUR RESPONSE: We thank the respected reviewer #2 for your useful comment, and we have revised accordingly.
Reviewer 3 Report
The authors performed a comparative transcriptome analysis to establish a lncRNA-mRNA network in rice infected by RBSDV. A total of 1342 mRNAs and 22 lncRNAs were found to be differentially-expressed after RBSDV infection. In the network, 56 plant-pathogen interaction-related DEmRNAs were co-expressed with 21 DElncRNAs, suggesting their roles in rice innate immunity against RBSDV. The lncRNA-mRNA regulatory relationships were experimentally verified in rice calli by an Agrobacterium-mediated transformation method. Five mRNAs were found regulated by 3 DElncRNAs. This information allows us to focus on candidate genes responsible for RBSDV infection in rice for further studies. Experimental design and results were appropriate and the conclusion made was sound.
Author Response
Reviewer: #3
Comments and Suggestions for Authors
The authors performed a comparative transcriptome analysis to establish a lncRNA-mRNA network in rice infected by RBSDV. A total of 1342 mRNAs and 22 lncRNAs were found to be differentially-expressed after RBSDV infection. In the network, 56 plant-pathogen interaction-related DEmRNAs were co-expressed with 20 DElncRNAs, suggesting their roles in rice innate immunity against RBSDV. The lncRNA-mRNA regulatory relationships were experimentally verified in rice calli by an Agrobacterium-mediated transformation method. Five mRNAs were found regulated by 3 DElncRNAs. This information allows us to focus on candidate genes responsible for RBSDV infection in rice for further studies. Experimental design and results were appropriate and the conclusion made was sound.
OUR RESPONSE: We would like to thank the respected reviewer #3 for your positive comments.
Round 2
Reviewer 1 Report
In the revised version by Zhang et. al., the manuscript has been substantially improved. Mainly, the relevance of the RNA-seq results has been correlated with RBSDV biology. For example, ROS, ubiquitin-proteosome, and hormone signaling has been described as it pertains to RBSDV biology and the differential gene expression observed during infection. The figures have been improved by increasing their size, although the panel labels were inadvertently increased in size for select panels (Figs. 1 and 3). Collectively, this study lays the groundwork for future studies on whether lncRNAs regulate mRNAs during RBSDV infection to either enhance or decrease virus fitness.
The authors have stated that they have presented the heatmaps, networks, etc. as an effort to support future studies by additional groups. The RNA-seq data, especially the annotated lncRNAs should be deposited into a database for easy distribution. Has the RNA-seq data and annotated lncRNAs been deposited to something like GEO omnibus?
If the data has not been deposited, how can this data be of use to others? The sequences of the lncRNAs should be available to others or the data presented in all of the heatmaps, networks etc. can not be studied further. Have the lncRNA sequences been described elsewhere? If so, this should be made clear and referenced.
Author Response
Reviewer: #1
Comments and Suggestions for Authors
In the revised version by Zhang et. al., the manuscript has been substantially improved. Mainly, the relevance of the RNA-seq results has been correlated with RBSDV biology. For example, ROS, ubiquitin-proteosome, and hormone signaling has been described as it pertains to RBSDV biology and the differential gene expression observed during infection. The figures have been improved by increasing their size, although the panel labels were inadvertently increased in size for select panels (Figs. 1 and 3). Collectively, this study lays the groundwork for future studies on whether lncRNAs regulate mRNAs during RBSDV infection to either enhance or decrease virus fitness.
OUR RESPONSE: We would like to thank the respected reviewer #1 for your positive comments.
The authors have stated that they have presented the heatmaps, networks, etc. as an effort to support future studies by additional groups. The RNA-seq data, especially the annotated lncRNAs should be deposited into a database for easy distribution. Has the RNA-seq data and annotated lncRNAs been deposited to something like GEO omnibus?If the data has not been deposited, how can this data be of use to others? The sequences of the lncRNAs should be available to others or the data presented in all of the heatmaps, networks etc. can not be studied further. Have the lncRNA sequences been described elsewhere? If so, this should be made clear and referenced.
OUR RESPONSE: We would like to thank the respected reviewer #1 for your good comment. According to your comment, we have deposited the RNA-seq data in SRA database (SRA accession: PRJNA657713), and uploaded the data of transcriptome analysis including the annotated lncRNAs and the data of Fragments per kilo-base per million reads (FPKM) to GEO database. We think the deposited data will be useful to other researches who is intended to mine additional information for his/her study.